# Korean Red Ginseng Ameliorates Fatigue via Modulation of 5-HT and Corticosterone in a Sleep-Deprived Mouse Model

**DOI:** 10.3390/nu13093121

**Published:** 2021-09-06

**Authors:** Ji-Yun Kang, Do-Young Kim, Jin-Seok Lee, Seung-Ju Hwang, Geon-Ho Kim, Sun-Hee Hyun, Chang-Gue Son

**Affiliations:** 1Institute of Bioscience & Integrative Medicine, Daejeon Oriental Hospital of Daejeon University, 75, Daedeok-daero 176, Seo-gu, Daejeon 35235, Korea; kangjy0118@naver.com (J.-Y.K.); neptune@dju.kr (J.-S.L.); bluesea9292@naver.com (S.-J.H.); 2Department of Korean Medicine, Korean Medical College of Daejeon University, 62, Daehak-ro, Dong-gu, Daejeon 34520, Korea; 95kent@naver.com (D.-Y.K.); kim5454ho@daum.net (G.-H.K.); 3R&D Headquarters, Korean Ginseng cooperation, Daejeon 34337, Korea; shhyun@kgc.co.kr

**Keywords:** central fatigue, chronic fatigue, corticosterone, Korean red ginseng, serotonin

## Abstract

Central fatigue, which is neuromuscular dysfunction associated with neurochemical alterations, is an important clinical issue related to pathologic fatigue. This study aimed to investigate the anti-central fatigue effect of Korean red ginseng (KRG) and its underlying mechanism. Male BALB/c mice (8 weeks old) were subjected to periodic sleep deprivation (SD) for 6 cycles (forced wakefulness for 2 days + 1 normal day per cycle). Simultaneously, the mice were administered KRG (0, 100, 200, or 400 mg/kg) or ascorbic acid (100 mg/kg). After all cycles, the rotarod and grip strength tests were performed, and then the changes regarding stress- and neurotransmitter-related parameters in serum and brain tissue were evaluated. Six cycles of SD notably deteriorated exercise performance in both the rotarod and grip strength tests, while KRG administration significantly ameliorated these alterations. KRG also significantly attenuated the SD-induced depletion of serum corticosterone. The levels of main neurotransmitters related to the sleep/wake cycle were markedly altered (serotonin was overproduced while dopamine levels were decreased) by SD, and KRG significantly attenuated these alterations through relevant molecules including brain-derived neurotropic factor and serotonin transporter. This study demonstrated the anti-fatigue effects of KRG in an SD mouse model, indicating the clinical relevance of KRG.

## 1. Introduction

Fatigue is both a physiological defense response and a disease-associated symptom; therefore, it is a common complaint in both the general population and patients with various disorders [1]. Fatigue can be generally classified according to duration as acute (≤1 month), prolonged (1< and ≤6 months), and chronic, lasting over 6 months [2]. Chronic fatigue is the main fatigue-related issue in the clinic, and its prevalence is approximately 10% in the general population [3]. In particular, medically unexplained chronic fatigue, such as chronic fatigue syndrome (CFS), has a more serious impact on health-related quality of life than brain stroke, angina pectoris, or schizophrenia [4].

Additionally, central fatigue is a neuromuscular dysfunction, a notable feature of chronic fatigue, that is caused by biochemical alterations in the brain [5]. Unlike peripheral fatigue, which is caused by energy-associated disturbances, mainly in muscles, central fatigue results from dysfunction of synaptic transmission in the central nervous system (CNS) [6]. Clinically, prolonged sleep disturbance and chronic stress are presumed to be inducers of central fatigue and are also major symptoms of pathologic central fatigue and chronic fatigue [7]. Important hypotheses on the pathophysiologic mechanisms of pathologic fatigue like CFS involve disruption of neuroendocrinological homeostasis resulting, for example, from chronic sleep deprivation (SD) [8].

Sleep is believed to play a key role in the maintenance of brain function and health, as well as in protection against and recovery from central fatigue [5,9]. Approximately 60% of subjects with chronic fatigue have comorbid sleep disorders, including insomnia, sleep apnea, and periodic limb movement disorder [10]. It is well known that chronic sleep restriction affects neuronal activity within the brain, altering serotonin (5-hydroxytryptamine, 5-HT) and dopamine (DA) levels, which are closely associated with the sleep/wake cycle [11]. While cortisol levels in the blood are elevated in the context of acute stress and general fatigue, this stress hormone is frequently depleted by prolonged SD and in subjects with severe central fatigue [12].

*Panax ginseng* (*P. ginseng*) is one of the most frequently employed herbs for various health issues, and it has shown moderate effects in treating fatigue [13,14]. Especially, Korean red ginseng (KRG), which is manufactured by repeated steaming and drying of raw ginseng, is known to exert pharmacologic activities, including antifatigue, antioxidative, and immunomodulatory effects [15,16]. To date, studies on the anti-fatigue effects of *P. ginseng* and KRG have focused on general or peripheral fatigue and have mainly involved measurement of energy metabolite- and oxidative stress-related marker levels in blood and tissues from clinical patients and animal models [17]. However, regarding central fatigue, evidence of the effects of *P. ginseng* and KRG is lacking.

To investigate the potential of KRG as a treatment for fatigue, we evaluated its efficacy and underlying mechanisms in a chronic sleep-deprived mouse model.

## 2. Materials and Methods

### 2.1. Materials

The following reagents were obtained from Sigma-Aldrich (St. Louis, MO, USA): a bicinchoninic acid protein assay (BCA) kit, ascorbic acid (AA), glycerol, 4′,6-diamidino-2-phenylindole dihydrochloride (DAPI), aqueous mounting buffer, sodium hydroxide, tetraethyl ethylenediamine (TEMED), copper (II) sulfate solution, sucrose, Tween 20, and bovine serum albumin (BSA). The following reagents were obtained from the following other manufacturers: Triton X-100; paraformaldehyde (PFA) powder, acetyl alcohol, hydrogen peroxide (H_2_O_2_), methylene alcohol, isopropanol, isopentene (Samchun Pure Chemical Co. Ltd., Seoul, Korea; Yakuri Pure Chemicals Co., Ltd., Kyoto, Japan); optimal cutting temperature (OCT) compound (Leica Microsystems, Bensheim, Germany); skim milk, 10% ammonium persulfate solution, RIPA buffer (LPS Solution, Daejeon, Korea); protease inhibitor, normal chicken serum, an Enhanced Chemiluminescence (ECL) Advanced Kit, antibodies against glucocorticoid receptor (GR), phospho-GR(p-GR), cAMP response element-binding protein (CREB), phospho-CREB (p-CREB), brain-derived neurotropic factor (BDNF), 5-HT, tryptophan hydroxylase 2 (TPH2), 5-HT transporter (5-HTT), 5-HT_1A_ receptor (5-HT_1A_R), tyrosine hydroxylase (TH), DA, β-actin, and α-tubulin, fluorescence- and horseradish peroxidase (HRP)-conjugated secondary antibodies (Novus, St. Louis, MO, USA; Abcam, Cambridge, MA, USA; Thermo-Fisher Scientific, Allentown, PA, USA; Cell Signaling Technology, Beverly, MA, USA); a corticosterone enzyme immunoassay kit (Arbor Assays Inc., Ann Arbor, MI, USA); RNA Later (Ambion, Austin, TX, USA); and polyvinylidene fluoride (PVDF) membranes (Pall Corporation, Port Washington, NY, USA).

### 2.2. Preparation and Fingerprinting of KRG

The KRG was manufactured and provided by the Korea Ginseng Corporation (KGC, Daejeon, Korea), and the fingerprinting analysis of KRG was conducted using an ultra-performance liquid chromatography with a photodiode array detector (UPLC-PDA) system (Waters Co., Milford, MA, USA), as described previously [18]. Reference peaks were detected after 13.64 (ginsenoside Rb1), 17.35 (Rg3s), 14.28 (Rc), 15.02 (Rb2), 17.37 (Rg3r), 16.04 (Rd), 11.76 (Rg2s), 9.09 (Rf), 11.51 (Rh1), 5.46 (Re), and 5.30 (Rg1) min of retention time. The concentrations of the reference compounds were 5.85 (ginsenoside Rb1), 4.43 (Rg3s), 2.29 (Rc), 2.17 (Rb2), 2.02 (Rg3r), 0.89 (Rd), 1.50 (Rg2s), 1.37 (Rf), 1.28 (Rh1), 0.82 (Re), and 0.69 (Rg1) mg/g KRG (Figure 1A).

### 2.3. Animals and Care

One hundred twenty-eight specific pathogen-free male BALB/c mice (8 weeks old, 21–24 g) were obtained from Dae Han Bio Link (Co., Ltd., Eumseong, Korea). The mice were given ad libitum access to water and food pellets (Cargill Agri Purina, Seongnam, Korea) and were housed in a room maintained at 23 ± 1 °C on a 12 h:12 h light–dark cycle. The animal care and experimental protocols were approved by the Institutional Animal Care and Use Committee of Daejeon University (DJUARB2020-028) and conducted in accordance with the Guide for the Care and Use of Laboratory Animals published by the U.S. National Institutes of Health (NIH). After acclimation for 7 days, the mice were used for experiments.

### 2.4. Apparatus for Inducing Sleep Deprivation

The modified multiple platform method (MMPM) was utilized to establish an animal SD model, as previously described [19]. Briefly, to induce chronic SD, the mice were placed in cages (27 × 42 × 18 cm) with multiple platforms (15 cylinders, each 3.5 cm in diameter) filled with tap water (23 ± 1 °C, 1 cm below the platform surface). They were able to move and sit on the platform but were not able to sleep because they fell down when they tried to sleep. The water was changed twice daily during the manipulation period. Food and water were provided through a grid placed on top of the water tank.

### 2.5. Experimental Design

Preliminary studies including modeling of fatigued mouse by using the MMPM and assessing the pharmacological effects of KRG were conducted. To generate a chronic fatigue model in the preliminary study, we subjected mice (12 total) to SD for 48 h using the MMPM and then allowed them to rest for 24 h in a normal home cage (*n* = 6). We confirmed that compared with normal mice, the SD-exposed mice exhibited fatigue-like behavior after the 2nd, 4th, and 6th SD using the rotarod test (*n* = 6, Figure 1B and Figure 2A). We also observed the antifatigue effects of KRG (100 and 400 mg/kg) and AA (100 mg/kg), as a positive control compound (*n* = 5 per group, 20 mice total), under normal conditions (Figure 2B). Based on the effects of these clinical doses, we chose the final experimental dose of KRG.

For the main experiment, mice (96 total) were randomly divided into 6 groups (*n* = 16 each) and orally administered water (normal, SD), KRG (100, 200, or 400 mg/kg) or AA (100 mg/kg) once daily at 10:00~11:00 a.m. throughout the 6 SD cycles. These mice (except those in the normal group) were subjected to SD for 6 cycles; half of the mice in each group were used for the two behavior tests, and the rest of the mice were used for blood collection and brain tissue sampling. On the last day of the experiment (during the rest period of the 6th cycle), the anti-fatigue effects of the treatments were evaluated using both the grip strength test (at approximately 11:00 a.m.) and rotarod test (at approximately 11:00 p.m.) (Figure 1B).

### 2.6. Rotarod Test

Fatigue-like behaviors were evaluated using a rotarod apparatus (ENV-574M, Med Associates Inc., St. Albans, VT, USA) according to the manufacturer’s instructions. After acclimation for 30 min in the testing room, the mice were trained to stay on a rotating rod (2–20 rpm) for 500 s. The mice were immediately placed back on the rod if they fell off during habituation. After 30 min, fatigue-like behaviors were evaluated (3 trials, interval of 15 min) as the speed of the rod was accelerated from 4–40 rpm. In this period, the latency to fall was recorded.

### 2.7. Grip Strength Test

Muscle strength was assessed by using a grip strength apparatus (BIO-G53, BIOSEB, Pinellas Park, FL, USA) connected to a wire grid (9 cm × 15 cm) and an isometric force transducer. The mice were allowed to grasp the grid with their paws and were gently pulled backward until they released their grip within 3 s. Grip strength was evaluated (5 trials, interval of 5 min), and then the mean maximal force was expressed in Newtons (N).

### 2.8. Blood Collection and Brain Tissue Preparation

All mice were sacrificed under CO_2_ anesthesia immediately after the end of the last cycle. Blood was collected following the guidelines of the Institutional Animal Care and Use Committee. Serum was collected by centrifugation at 3000 rpm for 15 min at 4 °C and then stored at −80 °C. For immunofluorescence staining, after transcardial perfusion, the whole brains from 3 mice from each group were fixed in 4% PFA solution. The hypothalamus and raphe nucleus (RN) were isolated immediately from the whole brains of the remaining 5 mice, and then samples were stored at −80 °C or in RNA Later. Then, the hypothalamus and RN were isolated and homogenized in RIPA buffer for biochemical analyses, such as Western blotting. Protein concentrations were determined using a BCA protein assay kit by measuring the absorbance at 560 nm using a spectrophotometer (Molecular Devices Corp., Sunnyvale, CA, USA).

### 2.9. Determination of the Corticosterone Level in Serum

The serum level of corticosterone was measured using commercial enzyme-linked immunosorbent assay kits according to the manufacturer’s instructions (catalog no. K014-H5). Absorbance at 450 nm was measured using a UV spectrophotometer (Molecular Devices). Inter-assay variation was 10.3% and intra-assay variation was 4.3%. Sensitivity for the assay was approximately 17.5 pg/mL.

### 2.10. Western Blot Analysis

To evaluate the level of protein expression in the brain, the brain tissues were denatured by boiling for 10 min. Then, the samples were separated by 10% polyacrylamide gel electrophoresis and transferred onto PVDF membranes. After blocking in 5% skim milk for 1 h, the membranes were probed with primary antibodies such as GR (1:1000; MA1-510, Invitrogen), p-GR (1:1000; 4161s, Cell Signaling), CREB (1:1000; ab31387, Abcam), p-CREB (1:1000; ab32096, Abcam), BDNF (1:1000; ab108319, Abcam), 5-HT_1A_R (1:1000; ab85615, Abcam), 5-HTT (1:800; ab9726, Abcam), α-tubulin (1:1000; ab7291, Abcam), and β-actin (1:1000; MA5-11869, Thermo-Fisher Scientific) antibodies overnight at 4 °C. The membranes were washed 3 times and incubated with an HRP-conjugated anti-rabbit (1:5000; for p-CREB, CREB, BDNF, 5-HT_1A_R, and 5-HTT) or anti-mouse antibody (1:5000; for p-GR, GR, α-tubulin, and β-actin) for 45 min. The proteins were visualized using an ECL Advanced Kit. Protein expression was observed using the FUSION Solo System (Vilber Lourmat, Collegien, France), and band intensity was analyzed with ImageJ version 1.46 (NIH, Bethesda, MD, USA).

### 2.11. Immunofluorescence Staining

The brain tissues were immersed in 4% PFA solution for 72 h and subsequently cryoprotected in 10–30% sucrose solutions for 24 h each. The brain tissues were embedded in OCT compound with liquid nitrogen and cut into frozen coronal sections (35 μm) using a cryostat (CM3050S, Leica Microsystems, Nussloch, Germany). The frozen brain tissue sections were stored in cryoprotectant. To block endogenous peroxidase activity, the free-floating sections were immersed in 1% H_2_O_2_. The sections were treated with blocking buffer (5% normal chicken serum and 0.3% Triton X-100 in ice-cold PBS) and incubated with a 5-HT (1:200; ab66047, Abcam), TPH2 (1:200; NB100-74555, Novus), DA (1:100; NB110-2538, Novus), or TH (1:200; NB300-109, Novus) primary antibody overnight at 4 °C. After washing with ice-cold PBS, the sections were incubated with a goat anti-mouse IgG H&L (1:400; for DA; Alexa Fluor 488-conjugated; ab150129), donkey anti-goat IgG H&L (1:400; for 5-HT; Alexa Fluor 488-conjugated; ab150129), or goat anti-rabbit IgG H&L (1:400; for TH and TPH2; Alexa Fluor 488- and 594-conjugated; ab150077 and ab150080, respectively) secondary antibody for 2 h at room temperature. The 5-HT-, TPH2-, DA-, or TH-stained sections were subsequently exposed to DAPI (1:1000; D9542, Sigma-Aldrich). Immunoreactivity was observed under a fluorescence microscope (×71, Olympus, Tokyo, Japan), and the fluorescence intensity was quantified using ImageJ 1.46 software (NIH, Bethesda, MD, USA).

### 2.12. Statistical Analysis

The data are expressed as the means ± standard deviations. Statistical analysis was performed using GraphPad Prism 7 software (GraphPad, Inc., La Jolla, CA, USA). Statistical significance was determined by using one-way analysis of variance (ANOVA) followed by Dunnett’s test. In all analyses, *p* < 0.05 was considered significant.

## 3. Results

### 3.1. Development of an SD-Induced Fatigue Model

To confirm that the exercise performance of the SD model mice was impaired, we performed the rotarod test. Mice in the SD group fell off the rotarod significantly earlier than those in the normal group from the 1st test (0.76-fold during the 2nd cycle, *p* < 0.05) to the last cycle (0.7-fold during the 6th cycle, *p* < 0.05) (Figure 2A).

### 3.2. Effects of KRG on Fatigue-Like Behaviors

Based on the preliminary results (Figure 2B), we assessed the performance of the mice in the rotarod and grip strength test to evaluate the antifatigue effects of KRG. KRG treatment significantly attenuated SD-induced fatigue behaviors, improving exercise performance in the rotarod test (*p* < 0.05 and <0.01) and muscle strength in the grip strength test (*p* < 0.01 for 200 mg/kg; *p* < 0.05 for 400 mg/kg) (Figure 2C,D). No significant change in performance in either test was observed for the AA group.

### 3.3. Effects of KRG on the Levels of Corticosterone and/or Its Receptor in Serum and Brain Tissues

To examine the stress hormone-related response, we determined the levels of corticosterone in serum and its receptor in the hypothalamus. Six cycles of SD nearly completely depleted corticosterone in the blood (0.27-fold), but this depleting effect was significantly ameliorated by administration of KRG (*p* < 0.05 for 400 mg/kg; Figure 3A). The alteration in corticosterone receptor expression in the hypothalamus was also significantly attenuated in the KRG-treated group compared with the SD group, as determined by the p-GR/GR ratio (Figure 3B,C). AA treatment did not induce significant changes in corticosterone levels or the p-GR/GR ratio.

### 3.4. Effects of KRG on the CREB–BDNF Pathway in the Brain

To evaluate neuronal activity in the hypothalamus, we measured the protein levels of CREB, p-CREB, and BDNF. There was a notable reduction in the p-CREB/CREB ratio (0.58-fold) and the protein level of BDNF (0.66-fold) in the SD group compared with the normal group, whereas these reductions were significantly reversed by administration of KRG (*p* < 0.01 for the p-CREB/CREB ratio for all doses; *p* < 0.01 for BDNF levels for 200 and 400 mg/kg; Figure 3B,C). AA had a similar effect as KRG.

### 3.5. Effects of KRG on the Levels of 5-HT, TPH2, and 5-HT Autoreceptors and Transporters in the Brain

To verify the regulatory effect of KRG on serotonergic activity, the levels of 5-HT-associated molecules in the RN were determined. SD induced overactivity of serotonin in the dorsal raphe nucleus (DRN), as evidenced by immunofluorescence staining for 5-HT (2.81-fold) and TPH2 (1.7-fold). The overactivation of both 5-HT (*p* < 0.01 for all doses) and TPH2 (*p* < 0.01 for all doses) was significantly attenuated in the KRG-treated group compared with the SD group (Figure 4A,B). Consistently, KRG markedly reversed the reduction in 5-HTT protein expression (*p* < 0.01 for all doses) but not 5-HT_1A_R protein expression (Figure 4C,D). AA treatment also exerted significant inhibitory effects against serotonergic hyperactivity. KRG treatment did not affect the p-GR/GR ratio, but AA increased the p-GR/GR ratio.

### 3.6. Effects of KRG on DA and TH Levels in the Brain

Dopaminergic activity was assessed by evaluating the effects of KRG on the levels of DA and TH (an enzyme responsible for DA synthesis) in the RN. DA (0.25-fold) and TH (0.29-fold) levels were notably decreased in the group exposed to six cycles of SD compared with the normal group, while these alterations in the levels of both DA (*p* < 0.01 for all doses) and TH (*p* < 0.01 for all doses) in the DRN were significantly ameliorated by administration of KRG (Figure 3A,B). Consistent with the histological observations, the SD-induced decrease in TH protein levels was significantly reversed by KRG treatment (*p* < 0.01; Figure 3C,D). These effects were similar to those observed in the AA-treated group.

## 4. Discussion

In this study, we investigated the anti-fatigue effects of KRG and obtained positive results, as described above. We subjected mice to chronic SD to induce hyperactivity of 5-HT in the brain, which is one of widely accepted mechanisms of central fatigue [5]. Many clinical studies have reported 5-HT hyperactivity in the brains of subjects suffering from severe fatigue, such as CFS [20,21]. One group demonstrated that injection of a 5-HT precursor into the brain could lead to exhaustive motor fatigue [22].

We tried to mimic the clinical environment, especially the sleep and rest insufficiency of modern lifestyles, increased behavioral lethargy, and neuroendocrinological dysregulation [23]. As expected, 6 cycles of periodic SD caused impaired exercise performance and induced an approximately 3-fold increase in 5-HT levels in the RN (Figure 2A and Figure 4B). In addition, we verified that corticosterone was depleted in the serum (Figure 3A). These three alterations (lethargic behavior, 5-HT hyperactivity, and decreased levels of corticosterone) are typical features of severe central chronic fatigue, likely CFS [8]. In general, corticosterone, a key modulator of the stress response, is well known to play a critical role in recovery from fatigue and to contribute to circadian rhythm maintenance by regulating neurotransmitters in the brain [24,25].

Impaired exercise capability is a common complaint of subjects with chronic fatigue. In particular, sufferers of CFS are vulnerable to becoming fatigued after performing daily tasks in the home without exhibiting any muscular defects, which is referred to as postexertional malaise (PEM) [26]. The low force generated by motor units in CFS subjects is known to result from muscle membrane dysfunction caused by central neural dysregulation [27]. The 5-HT hyperactivity in the CNS is the most well-known alteration in neurotransmitters that inhibits proprioceptive sensory and muscular contractions [28]. Ginseng improves physical function in both healthy individuals and subjects complaining of fatigue [14,29]. Likewise, in the present study, KRG improved exercise performance in the rotarod test under both normal conditions and after SD (Figure 2B,C). Rotarod performance reflects integrated motor functions, including muscle endurance, balance, and coordination [30].

The activity of 5-HT in the brain is strongly and positively correlated with a lack of sleep [31]. A human study revealed that a single day of SD elevated the plasma level of tryptophan, a precursor of 5-HT, by 20% [32]. As expected, 6 cycles of SD markedly elevated the 5-HT level and that of its synthetase (TPH2), but these alterations were significantly normalized by KRG treatment (Figure 4A,B). In the context of chronic SD, overproduced 5-HT spills into the extrasynaptic space and activates 5HT_1A_R (a representative inhibitory receptor) on the initial axon segments of motor neurons, leading to inhibition of neuronal output [6]. The 5-HT level is controlled by the balance between its release and reuptake, which are determined by the p-GR/GR ratio (desensitizing 5HT_1A_R results in release and activation) and the expression of its reuptake transporter (5-HTT) in the RN [33,34]. Six cycles of SD increased the p-GR/GR ratio without changing the expression of 5HT_1A_R in the RN (Figure 4C,D). A large amount of previous data showed that 5HT_1A_R desensitization (loss of the inhibitory function of 5-HT release) usually occurs without a change in the protein level of 5HT_1A_R, as shown in our results [33]. KRG treatment did not affect the release of 5-HT (GR activity or HT1AR expression) but increased its reuptake (increase in 5-HTT level) (Figure 4C,D). This suggests that administration of KRG might regulate serotonergic activity by promoting the reuptake of 5-HT in the synaptic space and that this may be the main mechanism of its anti-central fatigue properties.

Sufferers of chronic fatigue rarely experience restorative sleep and exhibit disrupted neuroendocrinological homeostasis [8]. A total of 87–95% of patients with CFS experience unrefreshing sleep [35]. Prolonged sleep disturbance dysregulates the hypothalamic–pituitary–adrenal (HPA) axis, especially in the metabolism of cortisol, which plays an important role in the stress response and the formation of sleep architecture [36]. Unlike acute stress, chronic stress induces low levels of serum cortisol and dysfunction of GR dysfunction in the hypothalamus by impeding its translocation and phosphorylation, resulting in a condition that is sometimes called adrenal fatigue [37]. In our SD model, serum corticosterone was almost completely depleted, and the GR ratio in the hypothalamus was increased; however, KRG treatment significantly attenuated these alterations in serum corticosterone levels and the p-GR/GR ratio (Figure 3A,C). Several clinical studies have shown that serum corticosterone levels can be normalized by therapeutic agents, resulting in positive outcomes for patients with chronic fatigue [38]. The hypothalamus is a central region that responds to stress, sleep, and fatigue, and hypothalamic neurons can be damaged under extremely unfavorable conditions [39]. BDNF supports neuronal survival, growth, and differentiation and may indicate the health status of the hypothalamus [40]. In our study, KRG treatment significantly attenuated the SD-induced downregulation of the expression of both BDNF and its transcription factor (as indicated by the p-CREB/CREB ratio) (Figure 3B,C).

In modern society, it is common for individuals to have trouble sleeping in terms of both quantitative and qualitative parameters, as approximately 8–18% of the general population worldwide experience such issues [41]. An increase in 5-HT levels within the CNS resulting from sleep disturbance acts as a homeostatic pressure that causes symptoms such as drowsiness, lethargy, and loss of strength, leading the individual to rest [31]. In contrast with decreased 5-HT levels, decreased DA levels in the RN under chronic SD conditions lead to less activity and less arousal [42]. In our study, KRG treatment reversed the SD-induced reduction in the levels of DA and TH, which are enzymes involved in DA synthesis, in the RN (Figure 5), indicating that KRG balanced the levels of these neurochemicals. Some RCTs have proven that AA exerts antifatigue effects [43,44], so we used it as a positive control in this study. Treatment with 100 mg/kg AA modulated the levels of 5-HT (resulting in a much lower level than that induced by any dose of KRG, *p* < 0.01, data not shown) and DA (exhibiting a similar effect as KRG); however, we did not observe improvements in exercise performance or corticosterone levels in the blood (Figure 2C,D and Figure 3A). Presumably, exercise performance was not improved because the 5-HT level was 2-fold higher than that under normal conditions, and the changes in corticosterone levels and GR function in the hypothalamus were not significantly ameliorated (Figure 4B,D). Considering the results obtained for AA, further pathophysiological studies on 5-HT- and corticosterone-related mechanisms of central fatigue are needed.

No conventional therapeutics for chronic fatigue are available yet; thus, many nonresponders to rest use complementary therapies, including herbal medicines [45]. Traditionally, *P. ginseng* and KRG have been used as tonics for increasing vital energy and recovering from fatigue in eastern Asia. A few studies revealed the anti-fatigue effects of KRG by clinical (180 Chinese participants or 50 Korean patients) and preclinical (mice) investigation; however, their focus of anti-fatigue effect was not based on neurobiological underlying mechanism [46,47,48]. The present study is the first evidence for the anti-fatigue effect of KRG focusing on neurochemical alterations related to fatigue. The dose of KRG used in the present study was chosen based on the dose commonly used in the clinic, i.e., 2 g in adults. As limitations, KRG generally contains more ginsenosides, including Rg3, than *P. ginseng* [49]. We currently do not know which components of KRG are mainly responsible for its effects and whether it has similar effects on severe central fatigue in humans, such as in those suffering from CFS. Additionally, owing to the mechanism of central fatigue related to CNS factors, precise electrophysiological and micro histological studies are needed to support whether motor dysfunctions occurred due to peripheral factors such as muscular defects.

Taken together, our results indicate that KRG can protect against or alleviate fatigue related to chronic SD which could contribute to central fatigue. The underlying mechanisms may involve the modulation of neurotransmitters, including serotonin in the brain, and corticosterone. Our data suggest that the therapeutic potential of KRG for patients suffering from severe fatigue should be evaluated clinically in the future.

## Figures and Tables

**Figure 1 nutrients-13-03121-f001:**
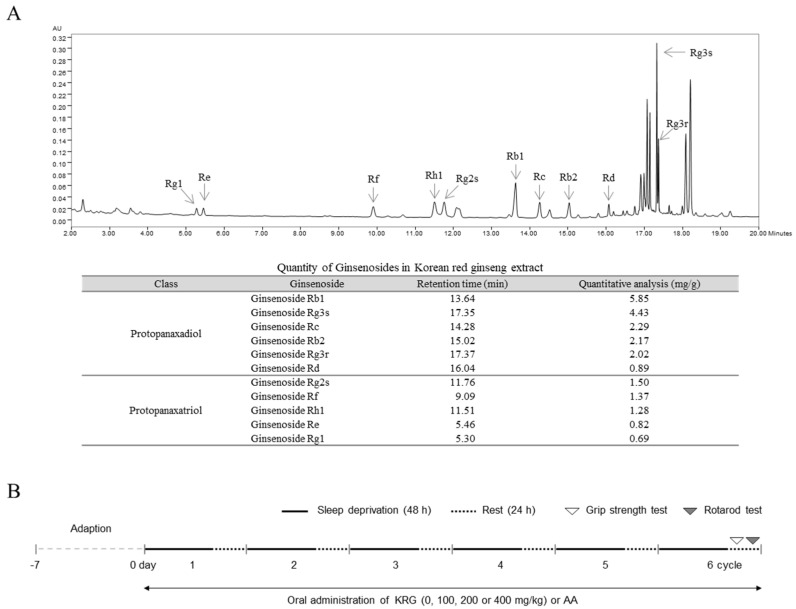
Fingerprinting analysis of KRG. KRG was subjected to UPLC-PDA analysis, and a chromatogram was obtained at a UV wavelength of 203 nm. The contents of different components of KRG, including protopanaxadiol (Rb1, Rg3s, Rc, Rb2, Rg3r, and Rd) and protopanaxatriol (Rg2s, Rf, Rh1, Re, and Rg1), were quantified (**A**). The diagram shows the experimental design used in this study (**B**).

**Figure 2 nutrients-13-03121-f002:**
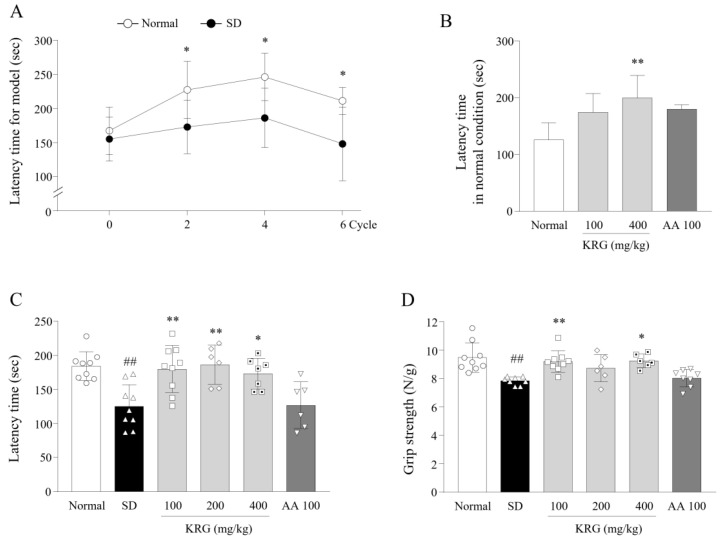
Effects of KRG on performance in the rotarod test and grip strength test. The rotarod test was conducted to assess the exercise capacity of the mice. The tests were used to verify that the SD model mice exhibited fatigue-like behavior (**A**) and to assess the effect of KRG on exercise performance under normal conditions (**B**), and after 6 cycles of SD (**C**). To investigate the effect of KRG on muscle strength, the grip strength test was performed at the end of the last SD cycle (**D**). The data are expressed as the means ± standard deviations (*n* = 5–8). ##, *p* < 0.01 compared with the normal group; *, *p* < 0.05 and **, *p* < 0.01 compared with the SD group.

**Figure 3 nutrients-13-03121-f003:**
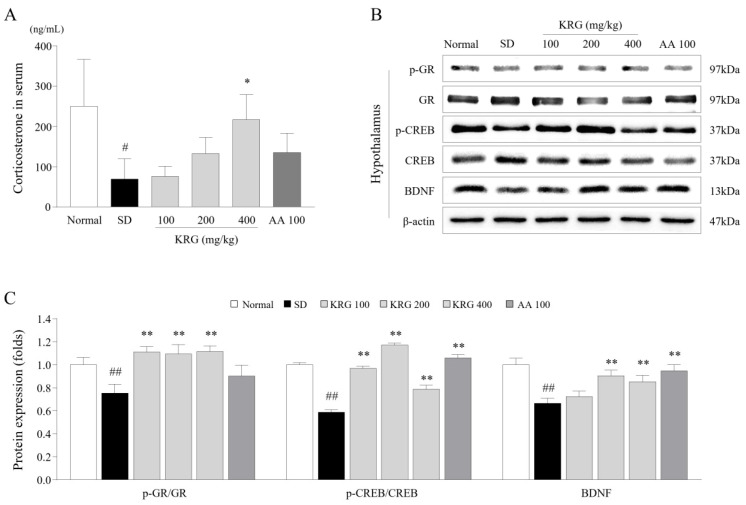
Effects of KRG on the levels of corticosterone and its receptors and the CREB–BDNF pathway in the serum and hypothalamus. The level of corticosterone in serum was determined using ELISA (**A**). Western blot analysis was used to evaluate the expression of GR- and BDNF-related proteins (**B**). The p-GR/GR ratio, p-CREB/CREB ratio, and BDNF protein level were semiquantified (**C**). The data are expressed as the means ± standard deviations (*n* = 5). #, *p* < 0.05 and ##, *p* < 0.01 compared with the normal group; *, *p* < 0.05 and **, *p* < 0.01 compared with the SD group.

**Figure 4 nutrients-13-03121-f004:**
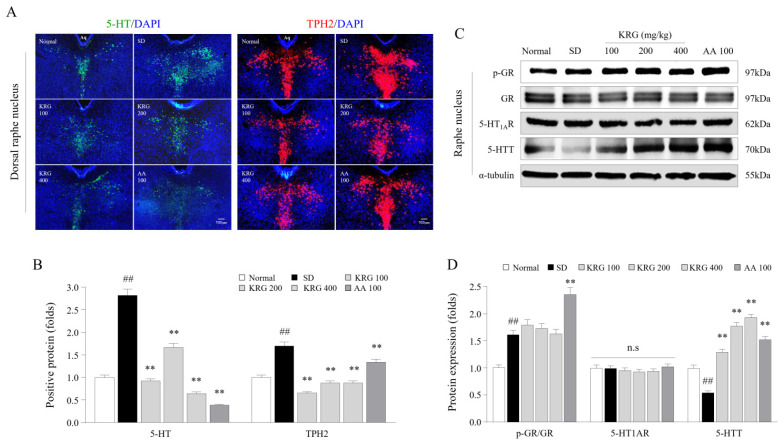
Effects of KRG on serotonergic activity-related molecules in the raphe nucleus. The relative 5-HT and TPH2 levels in the DRN were measured using immunofluorescence (**A**), and the relative 5-HT and TPH2 levels were semiquantified (**B**) (*n* = 3). Representative photomicrographs were taken at a magnification of 100×. Western blot analysis was used to evaluate p-GR, GR, 5-HT_1A_R, and 5-HTT protein levels in the RN (**C**), which were semiquantified (**D**) (*n* = 5). The data are expressed as the means ± standard deviations (*n* = 3 or 5). ##, *p* < 0.01 compared with the normal group; **, *p* < 0.01 compared with the SD group.

**Figure 5 nutrients-13-03121-f005:**
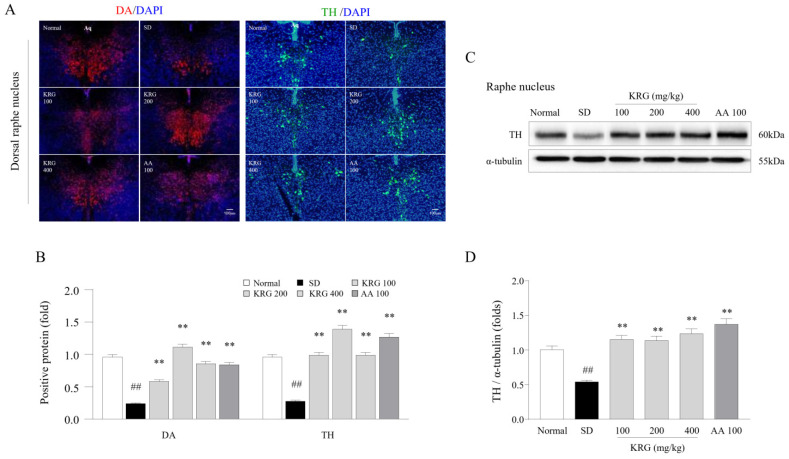
Effects of KRG on the levels of dopaminergic activity-related molecules in the raphe nucleus. The relative DA and TH levels in the DRN were measured using immunofluorescence (**A**), and the relative DA and TH levels were semiquantified (**B**) (*n* = 3). Representative photomicrographs were taken at a magnification of 100×. Western blot analysis was used to evaluate TH protein levels in the RN (**C**), which were semiquantified (**D**) (*n* = 5). The data are expressed as the means ± standard deviations (*n* = 3 or 5). ##, *p* < 0.01 compared with the normal group; **, *p* < 0.01 compared with the SD group.

## Data Availability

The original contributions presented in the study are included in the article/supplementary material; further inquiries can be directed to the Chang-Gue Son (corresponding author).

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
