# Peer review of "Korean Red Ginseng Ameliorates Fatigue via Modulation of 5-HT and Corticosterone in a Sleep-Deprived Mouse Model"

_nutrients, 2021, doi:10.3390/nu13093121_

Round 1

Reviewer 1 Report

This paper addressed the beneficial effect of Korean red ginseng (KRG) on central fatigue by using a chronic sleep-deprived mouse model.

In this study, authors use in vivo (rotarod and grip strength test) ed ex vivo approaches (ELISA, Immunofluorescence staining, western blot analysis) showing that KRG protects against or alleviated chronic central fatigue and underlying its clinical relevance.

I don't have any major criticisms regarding the study's results and conclusions. The data are clear and well presented. 

I have only a request. In ELISA experiments authors have to indicate the intra‐ and inter‐assay coefficients of variations and the assay's sensitivity.

Minor comments:

Figures 1A and 1B need a major resolution

Figures 4 and 5: the immunofluorescence is not clear. I suggest dividing the figures so that you can insert larger images especially with regards to immunofluorescence

line 261: add "the second A" in "A had similar..."

Author Response

[Reviewer 1]

This paper addressed the beneficial effect of Korean red ginseng (KRG) on central fatigue by using a chronic sleep-deprived mouse model.

In this study, authors use in vivo (rotarod and grip strength test) ed ex vivo approaches (ELISA, Immunofluorescence staining, western blot analysis) showing that KRG protects against or alleviated chronic central fatigue and underlying its clinical relevance.

I don't have any major criticisms regarding the study's results and conclusions. The data are clear and well presented. 

  1. I have only a request. In ELISA experiments authors have to indicate the intra‐ and inter‐assay coefficients of variations and the assay's sensitivity.

▶ We sincerely appreciate reviewer for the helpful request. We add information about intra‐ and inter‐assay coefficients of variations and the assay's sensitivity. (line 173-174)

  1. Figures 1A and 1B need a major resolution

▶ We really thank reviewer for the pragmatic comment. We adjusted resolution of the Figures for readers’ convenience.

  1. Figures 4 and 5: the immunofluorescence is not clear. I suggest dividing the figures so that you can insert larger images especially with regards to immunofluorescence

▶ We sincerely appreciate for the helpful suggestion. We deeply agreed with the unclearness of the Figures, so we increased the size and resolution of the immunofluorescence images instead of dividing Figures.

  1. line 261: add "the second A" in "A had similar..."

▶ We really thank for the detail review. We corrected the typo by adding “A” meaning ascorbic acid. (line 262)

Reviewer 2 Report

Korean red ginseng ameliorates central fatigue via modulation 2 of 5-HT and corticosterone in a sleep-deprived mouse model

The manuscript is related to very interesting issue of central fatigue animal model which translates to human major problem with sleep deprivation fatigue.  

There are only two minor remarks which need to be addressed.

1. Introduction section: heart angina  should be named either angina or angina pectoris even though  heart angina is present in the cited article it does not seem right similarly as stroke brain ( it should be brain stroke) and  diabetes type II  ( it should be diabetes type 2)present in the cited article.

2. Line 119-120 - the sentence needs grammar correction

Author Response

[Reviewer 2]

The manuscript is related to very interesting issue of central fatigue animal model which translates to human major problem with sleep deprivation fatigue.  

There are only two minor remarks which need to be addressed.

  1. Introduction section: heart angina should be named either angina or angina pectoris even though heart angina is present in the cited article it does not seem right similarly as stroke brain (it should be brain stroke) and diabetes type II (it should be diabetes type 2) present in the cited article.

▶ We sincerely appreciate reviewer for the critical and helpful comment. We changed the term to “angina pectoris” (line 38).

  1. Line 119-120 - the sentence needs grammar correction

▶ We really thank for detail review. We revised the sentence by correcting grammar mistakes (line 119-120)
